# Multi-Label Feature Selection for High-Dimensional Biological Data via Global Relevance and Redundancy Optimization based on JS Divergence

Man Yang
Institute of Data and Knowledge
Engineering, School of Computer and
Information Engineering,
Henan University,
Kaifeng, 475004, China
yangman@henu.edu.cn

Yibo Wang
Institute of Data and Knowledge
Engineering, School of Computer and
Information Engineering,
Henan University,
Kaifeng, 475004, China
15137005528@163.com

Yadi Wang*
Institute of Data and Knowledge
Engineering, School of Computer and
Information Engineering,
Henan University,
Kaifeng, 475004, China
yadiwang@henu.edu.cn

Xiaoding Guo
Institute of Data and Knowledge
Engineering, School of Computer and
Information Engineering,
Henan University,
Kaifeng, 475004, China
gxd@henu.edu.cn

Huiyu Mu
Institute of Data and Knowledge
Engineering, School of Computer and
Information Engineering,
Henan University,
Kaifeng, 475004, China
muhy@henu.edu.cn

Hangjun Che
College of Electronic and Information
Engineering, Southwest University,
Chongqing 400715, China
hjche123@swu.edu.cn

*Abstract*—In recent years, multi-label feature selection has been widely used in fields such as bioinformatics, information retrieval, and multimedia annotation. As an effective data pre-processing step, multi-label feature selection has shown its effectiveness in dealing with high-dimensional biological data in fields such as bioinformatics. Most of the previous multi-label feature selection methods are directly transformed from the traditional single-label feature selection methods, or they cannot make full use of label information. As a result, the selected feature subset involves features that are redundant or irrelevant to label information. Moreover, most algorithms do not use discretization when processing continuous data sets, so they cannot effectively eliminate the interference of abnormal data. In order to solve these problems, based on the GRRO (Global Relevance and Redundancy Optimization) algorithm introduces JS divergence to measure the correlation between labels and introduces data discretization pre-processing operations for continuous data sets. The experimental results after experimental verification on ten typical high-dimensional biological data sets show that the GRRO-JS algorithm is superior to the traditional multi-label feature selection method and the GRRO algorithm in terms of accuracy and efficiency and has high practical value.

*Index Terms*—multi-label feature selection, high-dimensional biological data, JS divergence, data discretization

This work is supported by grants from the National Natural Science Foundation of China (Nos. 62106066), the Henan Province Science Foundation of Excellent Young Scholars (242300421171), the Henan Province Science Foundation of Young Scholars (242300421703), the Key Research Projects of Henan Higher Education Institutions (No. 22A520019), the Key Research and Promotion Projects in Henan Province under Grant (No. 232102210080).

## I. INTRODUCTION

In traditional single-label supervised learning, each instance is associated with only one class label. However, in real-world scenarios, instances are often associated with multiple semantics [1]. For example, in bioinformatics, a single gene can be associated with multiple labels; in information retrieval, each document can cover multiple topics; and in image recognition, different scenes can be used to annotate images, resulting in multiple categories. Like single-label learning, multi-label data often contain thousands or even tens of thousands of features [2]. Given a specific learning task, many of these features may be redundant or irrelevant, and high-dimensional data can lead to computational inefficiency, overfitting, and poor performance in learning models [3]. To address this issue, researchers have developed multi-label feature selection algorithms to reduce the dimensionality of multi-label data, improve classification accuracy, and generate more compact and generalizable learning models. As a result, multi-label feature selection has become an important research topic and a current research hotspot. Nowadays, multi-label feature selection has been widely applied in fields such as bioinformatics [4], image annotation [5], and video classification [6]. For instance, in biomedical fields, multi-label feature selection is widely used in case data analysis to extract various information related to cancer, thereby improving the cure rate for cancer.

In recent years, a large number of multi-label feature selection methods have been proposed. Fan et al. [7] developed an algorithm called "Multi-label feature selection with local

discriminant model and label correlations." This method takes into account the neighboring instances of a given instance, constructs local clusters for the instances, and globally integrates local discriminant models to evaluate the clustering performance of all instances. Dai et al. [8] defined label gain based on label discriminant functions explored the correlations between features and proposed a multi-label feature selection method based on correlated label gain. He et al. [9] proposed a new method for label-enhanced multi-label feature selection based on sample correlation to explore the correlation between samples and labels. We proposed an improved method based on the GRRO algorithm [10] and JS divergence to address the problem of multi-label feature selection in high-dimensional biological data, aiming to enhance its accuracy and efficiency.

The main contributions of this paper can be summarized as follows:

- We propose the GRRO-JS algorithm, which improves the existing GRRO multi-label feature selection method by introducing the JS divergence to more effectively measure the correlation between labels. The JS divergence possesses ideal properties such as symmetry and non-negativity, making it particularly suitable for the multi-label feature selection process. By incorporating the JS divergence, we can more accurately capture and quantify the interdependence among labels, thereby enhancing the performance and precision of feature selection.
- We perform data discretization preprocessing on continuous datasets, converting continuous variables into discrete variables, which helps reduce data complexity and makes subsequent analysis more manageable. Additionally, we employ parallel computing techniques to measure the correlation between features, significantly reducing the algorithm's time complexity. This combination of data discretization and parallel computing not only improves the efficiency of the feature selection process but also enhances the algorithm's scalability, making it more suitable for handling high-dimensional datasets with many features.
- We conduct experiments on ten typical high-dimensional biological datasets, and the results demonstrate that GRRO-JS outperforms traditional multi-label feature selection methods and the original GRRO algorithm in terms of accuracy and efficiency. This improved performance highlights the algorithm's significant practical application value, especially in complex, high-dimensional data.

## II. PRELIMINARIES

### A. Multi-label Feature Selection

The goal of multi-label feature selection is to extract an optimal subset of features from the original features of the sample data. In multi-label classification, an instance can be associated with multiple labels. High-dimensional multi-label datasets not only lead to increased computational costs and storage requirements but also limit the practical use of machine learning models [11]. Feature selection has been proven to be highly effective in removing irrelevant and redundant features from the feature representation, thereby retaining the most discriminative information for multi-label learning. Therefore, it is crucial to design effective methods for multi-label feature selection.

Multi-label feature selection algorithms based on filter methods score each feature through statistical tests. This approach has a low computational cost and effectively avoids overfitting. Lin et al. [12] proposed the MDMR algorithm, which evaluates the importance of each feature based on neighborhood information entropy and uses a strategy of maximum dependency and minimum redundancy to obtain feature ranking. Lin et al. [13] proposed a multi-label feature selection algorithm based on neighborhood mutual information called MFNMI.

Multi-label feature selection algorithms based on wrapper methods use classifier performance as the search objective, continuously selecting and eliminating features until the optimal subset is found. This method is more targeted than filter methods and can improve model performance, but it also results in higher computational costs. Gharroudi et al. [14] proposed a wrapper-based feature selection method that considers label dependency and introduces a random forest-based multi-label wrapper feature selection method. Similarly, Zhang et al. [15] relied on principal component analysis and genetic algorithms to evaluate feature subsets.

Multi-label feature selection algorithms based on embedded methods integrate feature selection into the model training process. This approach uses certain feature evaluation properties inherent to the model as evaluation criteria and then applies a wrapper-based multi-label feature selection method to choose the optimal feature subset. However, this method can increase the training burden on the model. Zhang et al. [16] proposed an embedded multi-label feature selection method based on sparsity, which considers multiple regularizations and leverages local and global label correlations. Jian et al. [17] considered label correlations, reducing the dimensionality of the label space and using the selected label information for feature selection.

### B. GRRO Algorithm

The GRRO algorithm is based on an information-theoretic multi-label feature selection method. The optimization objective function of the GRRO algorithm is defined as follow:

$$
\min_{W} \ \|W - C\|_F^2 + \alpha tr\left(W^T G W\right) \\
+ \beta tr\left(R W^T W\right) \|W - C\|_F^2 \tag{1}
$$

it represents the squared Euclidean distance between the prediction matrix $W$ and the true label matrix $C$, which quantifies the difference between the predicted values and the true values. By minimizing the objective function, the GRRO algorithm aims to make the predictions as close as possible to the true labels. $\alpha tr\left(W^T G W\right) + \beta tr\left(R W^T W\right)$ is used to control the model's complexity and generalization ability. Here, $G$

and $R$ are positive semi-definite matrices. The notation $tr(\cdot)$ represents the trace of a matrix, which is the sum of its main diagonal elements. $a$ and $beta$ are weighting parameters that control the balance between data terms. By adjusting these parameters, the GRRO algorithm can avoid overfitting.

The GRRO algorithm has the following features: First, it takes into account the correlations between labels. In high-dimensional biological datasets, labels are often inter-dependent, making it crucial to consider label correlations. Second, the GRRO algorithm addresses feature redundancy, which helps avoid selecting duplicate or redundant features, thereby improving the efficiency and performance of multi-label feature selection. Additionally, the GRRO algorithm can handle high-dimensional datasets efficiently, as it only needs to traverse the correlation and redundancy information once to obtain the optimal solution easily.

### C. Jensen-Shannon Divergence

In probability theory or statistics, the JS divergence measures the distance between two probability distributions [18]. JS divergence is an improved method based on KL divergence that can completely resolve the asymmetry problem of KL divergence. Generally, given two discrete probability distributions $P$ and $Q$, the JS divergence is defined as: $JS(P\|Q) = \frac{1}{2}KL\left(P\left\|\frac{P+Q}{2}\right.\right) + \frac{1}{2}KL\left(Q\left\|\frac{P+Q}{2}\right.\right)$. The range of JS divergence is $[0,1]$. For any two discrete probability distributions $P$ and $Q$, the more similar $P$ and $Q$ are, the smaller the JS divergence value; conversely, the more dissimilar $P$ and $Q$ are, the larger the JS divergence value.

In practical applications of multi-label feature selection, each label can be viewed as a discrete probability distribution, with its probability value representing the likelihood of that label appearing in a specific sample within the dataset. Using JS divergence for multi-label feature selection allows for a more comprehensive measurement of the correlations between labels, thereby improving the model's performance and efficiency. This approach aids in the more precise selection of feature subsets.

## III. METHOD

### A. CRRO-JS Algorithm

The GRRO algorithm works by finding an optimal feature matrix $W$, which ensures that the features correlate highly with the labels while minimizing redundancy between the features [10]:

$$\max_{W} \sum_{u=1}^{q} \sum_{i=1}^{d} \left( I\left(f_i, l_u\right) w_{iu} - \sum_{j=1}^{d} I\left(f_i, f_j\right) w_{iu} w_{ju} \right) \quad (2)$$

where $W \in \mathbb{R}^{d \times c}$ represents the feature coefficient matrix, and $w_{iu} \in W$ indicates the importance of feature $f_i$ relative to label $l_u$. $I\left(f_i, l_u\right)$ and $I\left(f_i, f_j\right)$ represent the mutual information between feature $f_i$ and label $l_u$, and between feature $f_i$ and feature $f_j$, respectively.

Based on Eq. (2), the following optimization problem can be derived:

$$\min_{W} \|W - C\|_F^2 + \sum_{u=1}^{q} w_{.u}^T G w_{.u} \quad (3)$$

where $C$ represents the correlation between features and labels, with $c_{ij} \in C$ defined as $c_{ij} = I(f_i, l_j)$. $G$ represents the redundancy between features, with $g_{ij} \in G$ defined as $g_{ij} = I(f_i, f_j)$, and $G$ is a symmetric matrix. $\|\cdot\|_F$ denotes the Frobenius norm.

In Eq. 3, second-order label correlations are introduced to learn global label relationships, thereby improving the model's generalization performance.

$$\sum_{u=1}^{q} \sum_{j=1}^{q} r_{uj} w_{.u}^T w_{.j} \quad (4)$$

where $r_{uj}$ represents the correlation between label $l_u$ and label $l_j$. The GRRO algorithm directly calculates $r_{uj}$ using cosine distance.

For high-dimensional biological datasets, the correlations between labels are often more abstract and complex. To fully account for these correlations, we used JS divergence as a measure. We will discuss the advantages of JS divergence over cosine distance and mutual information in measuring label correlations.

For cosine distance, it only considers the similarity in direction between vectors without accounting for differences caused by other factors, such as the length of the vectors. This makes it unsuitable for measuring complex label correlations in high-dimensional biological datasets.

For mutual information, the calculation often includes some redundant information. This leads to inaccuracies when measuring the complex label correlations in high-dimensional biological datasets using mutual information. Additionally, since the calculation of mutual information relies on the joint probability distribution, it results in extremely high computational complexity when the number of labels is large.

In contrast, JS divergence can better capture the deviation information between labels in high-dimensional biological datasets compared to cosine distance, such as differences in the frequency of label occurrence in samples. On the other hand, JS divergence provides a more comprehensive measure of label correlations than mutual information. Additionally, since the calculation of JS divergence only requires considering the distribution information of each label in the dataset, it results in lower computational complexity.

Therefore, when measuring the correlations between label $l_u$ and label $l_j$, our GRRO-JS algorithm uses JS divergence for calculation. JS divergence is a particularly suitable method. Its ability to capture subtle differences in label distributions and its lower computational complexity makes it an effective tool for understanding and leveraging label correlations in complex, multi-label scenarios. This suitability is essential in contexts where accurately modelling these correlations can significantly enhance the performance and efficiency of feature selection algorithms.

Based on Eq. (3) and Eq. (4), the optimization objective function for GRRO can be derived. Since $\sum_{u=1}^{q} w_{.u}^{T} G w_{.u} = \sum_{u=1}^{q} (W^{T} G W)_{uu} = \text{tr}(W^{T} G W)$ and $\sum_{u=1}^{q} \sum_{j=1}^{q} r_{uj} w_{.u}^{T} w_{.j} = \text{tr}(R W^{T} W)$, where $R$ represents the relevance between labels. The optimization objective can be further transformed into:

$$\min_{W} \|W - C\|_{F}^{2} + \alpha \text{tr}(W^{T} G W) + \beta \text{tr}(R W^{T} W) \quad (5)$$

The feature coefficient matrix $W$ is included in all terms of Eq.(5), indicating that when optimizing the feature weight matrix using Eq. (5), the influences of feature relevance, feature redundancy, and label relevance can all be simultaneously considered. From Eq. (5), we know that the matrices $G$ and $R$ are semi-definite matrices. By setting the derivative of Eq. (5), with respect to $W$ to zero, we obtain:

$$2(W - C) + \alpha(G + G^{T})W + \beta W(R + R^{T}) = 0 \quad (6)$$

Since $G$ and $R$ are symmetric matrices, Eq. (6) can be transformed into:

$$(I + \alpha G)W + \beta W R = C \quad (7)$$

where $I$ represents the identity matrix. Eq. (7) is a matrix equation of the form $AW + WB = C$, where $A = I + \alpha G$ and $B = \beta R$.

In MATLAB, the lyap function is used to solve linear matrix equations of the form $AW + WB + C = 0$, where $A$, $B$, and $C$ are known matrices, and $W$ is the matrix to be solved. Lyap function employs several numerical stability optimization techniques specifically designed to solve linear matrix equations of the form $AW + WB + C = 0$. Additionally, the lyap function offers a simple call syntax, requiring only the input matrices $A$, $B$, and $C$ to solve for matrix $W$. So, the importance of each feature can be determined based on the computed $\|w_{(i.)}\|_{2}$ (for $1 \leq i \leq d$).

*B. Pseudocode of the GRRO-JS Algorithm*

---
**Algorithm 1** Data Loading and Preprocessing
---
1: **Input:** High-dimensional biological multi-label classification data $M_{train}$ and $M_{text}$, with $F$ features, $L$ labels, and $N$ instances
2: function GRRO-JS
3: Load the features and labels of the training set $M_{train}$ and the test set $M_{test}$ for biological multi-label classification data
4: Set the parameters $\alpha$ and $\beta$
5: For continuous training data, determine the value of $numB$ based on feature values and discretize the continuous training data into $numB$ groups. This step is not required for binary training data.
---

In this paper, we introduce JS divergence into the GRRO algorithm to more accurately measure the correlation between labels. This approach aims to better adapt to multi-label classification problems in high-dimensional biological

datasets, enhancing the algorithm's performance in handling complex label relationships. By incorporating JS divergence, the GRRO-JS algorithm can more comprehensively capture the underlying connections between labels, thereby improving classification accuracy and efficiency. Algorithm 1 to Algorithm 6 present the detailed pseudocode of the GRRO-JS algorithm, providing a clear illustration of the core processes and implementation steps.

---
**Algorithm 2** Calculating the Correlation Between Features and Labels
---
1: **for** $i$=1:num_feature **do**
2:     **for** $j$=1:num_label **do**
3:         Calculate the mutual information between $F_i$ and $L_j$ as their correlation and store it in $FL$
4:     **end for**
5: **end for**
---

---
**Algorithm 3** Calculating the Redundancy Between Features
---
1: **for** $i$=1:num_feature **do**
2:     **for** $j$=1:num_feature **do**
3:         Calculate the mutual information between $F_i$ and $F_j$ as their redundancy and store it in $FF$
4:     **end for**
5: **end for**
---

---
**Algorithm 4** Calculating the Correlation Between Labels
---
1: **for** $i$=1:num_label **do**
2:     **for** $j$=1:num_label **do**
3:         Calculate the JS divergence between $L_i$ and $L_j$ as their correlation and store it in $LL$
4:     **end for**
5: **end for**
---

---
**Algorithm 5** Calculating the Weight Matrix $W$
---
1: $A = I + a * FF$
2: $B = \beta * LL$
3: $C = -FL$
4: $W = lyap(A, B, C)$
---

*C. Flowchart of the GRRO-JS Algorithm*

The workflow of the GRRO-JS algorithm begins with loading the high-dimensional biological dataset. The next step is to check whether the data type meets the required standard. If the data type is unsuitable and is not an integer, a discretization preprocessing step is necessary to convert the data into integers. If the data type is already appropriate, the algorithm sets the parameters for the optimization objective function.

Following this, mutual information is used to measure the relevance between features and labels (FL) and the redundancy between features (FF). Subsequently, JS divergence is employed to assess the correlations between labels (LL).

---

**Algorithm 6** Obtaining the Performance Metrics of the Classifier

---
1: **Output:** Performance metrics of the classifier
2: **for** $i$=1 to the top 50 features **do**
3:     Train the MLKNN classifier on the selected top $i$ features
4:     Calculate the six performance evaluation metrics
5:     Store the performance metrics
6: **end for**
7: Visualize the performance metrics
8: end function

---

The algorithm then uses the lyap function to calculate the weight matrix $W$ and rank the features accordingly. The top $i$ features are selected as the feature subset, which is then validated against six evaluation metrics to assess its performance.

The final steps involve saving the classification performance metrics obtained from the six evaluation metrics for the selected feature subset and terminating the function. Figure 1 illustrates the flowchart of the GRRO-JS algorithm, providing a visual representation of this process.

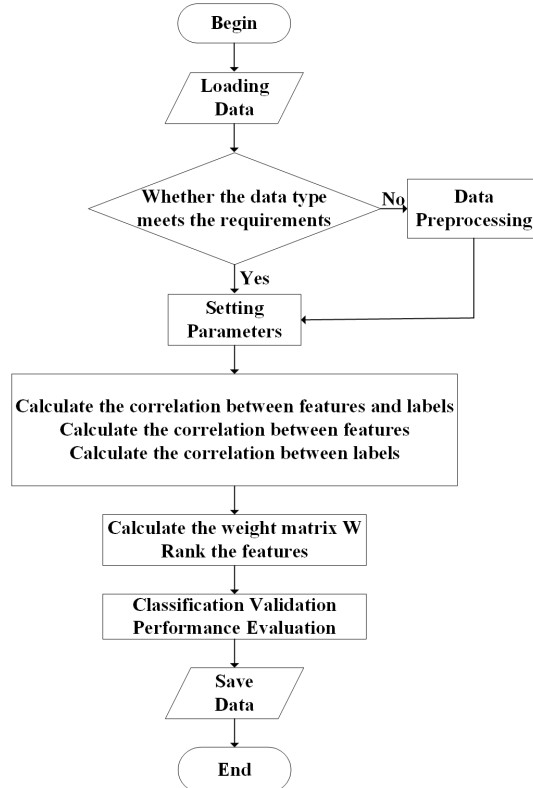

Fig. 1. Flowchart of the GRRO-JS algorithm.

## IV. EXPERIMENT

### A. Evaluation Metrics

We use six commonly employed multi-label evaluation metrics to assess performance, including four instance-based metrics: Hamming Loss, Ranking Loss, Coverage, and Average Precision, and two label-based metrics, Macro F1 and Micro F1.

Hamming Loss indicates the degree of disagreement between the predicted and actual label sets and the proportion of incorrect label predictions in the sample.

Ranking Loss calculates the number of labels in which the probability of a label being 1 is less than the probability of a label being 0, divided by the product of the number of labels being 1 and the number of labels being 0. This represents the ranking error in label predictions.

Coverage measures the extent to which labels with a value of 1 are covered in the ordered list of predicted labels, ranked from highest to lowest probability. It is the ratio of the union to the intersection of the predicted label set and the true label set. Average Precision indicates the average accuracy of the predicted label set.

Macro F1 considers the prediction for each label, first calculating the arithmetic mean of precision (the proportion of true positives among predicted positives) and recall (the proportion of true positives among actual positives) for each label. The harmonic mean of these two averages gives the Macro F1 score, which is the average F1 score across all labels.

Micro F1 considers the prediction for each sample, calculating the total precision and recall across all labels and then computing their harmonic mean to obtain the Micro F1 score. This is achieved by merging all prediction results into a single label set and calculating the F1 score for this combined set.

### B. Baselines

To compare the performance of the GRRO-JS algorithm with other algorithms, this experiment selected four additional multi-label feature selection methods to be compared across ten chosen high-dimensional biological multi-label datasets. These methods include a heuristic search algorithm PMU [1], a regularization-based multi-label feature selection method MDFS [19], a mutual information-based multi-label feature selection method D2F [20], and the GRRO algorithm. First, in this experiment, the top fifty features, determined by the sum of feature weights calculated by each algorithm, were selected for performance comparison between the GRRO-JS algorithm and the other four information theory-based algorithms. Then, the features ranked from 1 to 50 in each dataset were sequentially selected to further compare the performance of the GRRO-JS algorithm with that of the GRRO algorithm.

### C. Experimental Design

*1) Dataset:* In this experiment, we use ten high-dimensional multi-label biological datasets covering various categories such as organelles, viruses, plants, and humans, as well as different levels of biological information, including genes, pseudo-amino acid sequences, and proteins. Table I summarizes the detailed characteristics of these datasets.

TABLE I
THE STATISTICAL DATA OF THE TEN DATASETS.

| Datasets | Train | Valid | Test | Label |
|---|---|---|---|---|
| EukaryoteGO | 4658 | 3108 | 12689 | 22 |
| GnegativeGO | 836 | 556 | 1717 | 8 |
| GpositiveGO | 311 | 208 | 912 | 4 |
| HumanGO | 1862 | 1244 | 9844 | 14 |
| PlantGO | 588 | 390 | 3091 | 12 |
| VirusGO | 124 | 83 | 749 | 6 |
| EukaryotePseAAC | 4658 | 3108 | 440 | 22 |
| GnegativePseAAC | 836 | 556 | 440 | 8 |
| GpositivePseAAC | 311 | 208 | 440 | 4 |
| HumanPseAAC | 1862 | 1244 | 440 | 14 |

*2) Experimental Procedure:* Hyperparameter Settings: To control variables and ensure the validity of the analysis results, the GRRO-JS algorithm inherits the hyperparameters from the GRRO algorithm. When solving the feature weight matrix, two hyperparameters, and , need to be set. In the label classification after feature selection, the same ML-KNN classifier is used in this experiment, requiring the setting of the hyperparameter $k$. We use grid search to determine the hyperparameters.

Feature Selection: Feature selection consists of two parts: solving the feature matrix and sorting features. Before solving the feature matrix $W$, we first need to measure the correlation FL between features and labels, and the redundancy FF between features using the concept of mutual information. Then, the correlation LL between labels is measured using the concept of JS divergence. These three matrices are computed using nested loops. It is important to note that since the redundancy of a feature with itself is always 1, the diagonal elements of the matrix FF are explicitly set to 1. During the calculation, cases of missing labels or features can lead to matrix elements being unsolvable, resulting in 'nan' values. Therefore, 'nan' elements in the matrices are set to 0. Once FF, FL, and LL are obtained, the weight matrix $W$ can be calculated by solving the matrix equation of the form $AW + WB = C$, where $A = I + \alpha FF$, $I$ is the identity matrix, $B = \beta LL$, and $C = -FL$. Considering the data scale and equation form, the MATLAB function lyap is used to solve for the feature weight matrix $W$. Each column of the weight matrix $W$ represents the weight of different features for a single label, while each row represents a feature's weight across different labels. Summing $W$ row-wise gives the total weight of each feature across all labels, which is used as the basis for ranking the features.

Classification Validation: After obtaining the feature weight matrix, various classifiers can be used for classification, such as Random Forests, KNNs, Neural Networks, etc. To suit the characteristics of multi-label classification and variable control, this experiment continues to use the ML-KNN classifier from the GRRO algorithm and maintains the same hyperparameters.

Performance Evaluation: The performance evaluation function is defined within the MLKNN function. After running MLKNN, the six performance metricsHamming Loss, Ranking Loss, Coverage, Average Precision, Macro F1, and Micro

F1can be obtained.

*D. Experimental Results and Analysis*

*1) Comparison of the Performance of the GRRO-JS Algorithm with Other Algorithms:* Considering that different biological datasets contain distinct types of biological information and may have varying features and that an excessive number of features can add extra computational burden, this experiment selects the top 50 features based on the total feature weight computed by each algorithm for performance comparison. Tables II to VII present the results of the GRRO-JS algorithm and other algorithms across six evaluation metrics.

TABLE II
PERFORMANCE COMPARISON OF GRRO-JS ALGORITHM AND OTHER ALGORITHMS ON HAMMING LOSS

| Datasets | Hamming Loss | | | | |
| | GRRO-JS | GRRO | PMU | MDFS | D2F |
|---|---|---|---|---|---|
| EukaryoteGO | **0.019700** | **0.019700** | 0.022040 | 0.022289 | 0.026223 |
| GnegativeGO | **0.019784** | **0.019784** | 0.020683 | 0.026304 | 0.023381 |
| GpositiveGO | **0.042067** | **0.042067** | 0.049279 | 0.043269 | 0.044471 |
| HumanGO | 0.045475 | 0.044212 | 0.045590 | **0.042375** | 0.055409 |
| PlantGO | **0.040812** | **0.040812** | 0.041239 | 0.045513 | 0.043590 |
| VirusGo | **0.060241** | **0.060241** | 0.072289 | 0.066265 | 0.072289 |
| EukaryotePseAAC | 0.995452 | 0.997177 | 0.999079 | **0.993872** | 0.999313 |
| GnegativePseAAC | **0.976782** | 0.980551 | 0.982995 | 1.000000 | 0.981769 |
| GpositivePseAAC | 0.979240 | **0.976864** | 0.997596 | 0.989292 | 0.996941 |
| HumanPseAAC | 0.997442 | 0.996894 | 0.996967 | **0.996090** | 0.996711 |
| Average | **0.417700** | 0.417830 | 0.422776 | 0.422527 | 0.424010 |

The best results are highlighted in bold.

TABLE III
PERFORMANCE COMPARISON OF GRRO-JS ALGORITHM AND OTHER ALGORITHMS ON RANKING LOSS

| Datasets | Ranking Loss | | | | |
| | GRRO-JS | GRRO | PMU | MDFS | D2F |
|---|---|---|---|---|---|
| EukaryoteGO | **0.021496** | **0.021496** | 0.032066 | 0.032035 | 0.042475 |
| GnegativeGO | **0.011990** | **0.011990** | 0.018906 | 0.019848 | 0.020898 |
| GpositiveGO | 0.046474 | 0.046474 | 0.044471 | 0.041667 | **0.039663** |
| HumanGO | **0.037515** | 0.038101 | 0.049318 | 0.039543 | 0.067324 |
| PlantGO | 0.042100 | 0.042100 | **0.040386** | 0.051131 | 0.043032 |
| VirusGo | **0.034036** | **0.034036** | 0.048327 | 0.063253 | 0.048327 |
| EukaryotePseAAC | **0.118362** | 0.123835 | 0.132275 | 0.166631 | 0.132400 |
| GnegativePseAAC | **0.036241** | 0.036697 | 0.045992 | 0.117485 | 0.051702 |
| GpositivePseAAC | **0.030798** | 0.032171 | 0.043510 | 0.055523 | 0.043739 |
| HumanPseAAC | **0.100876** | 0.109055 | 0.112321 | 0.122744 | 0.114762 |
| Average | **0.047989** | 0.049596 | 0.056757 | 0.070986 | 0.060432 |

The best results are highlighted in bold.

TABLE IV
PERFORMANCE COMPARISON OF GRRO-JS ALGORITHM AND OTHER ALGORITHMS ON COVERAGE

| Datasets | Coverage | | | | |
| | GRRO-JS | GRRO | PMU | MDFS | D2F |
|---|---|---|---|---|---|
| EukaryoteGO | **0.631596** | **0.631596** | 0.888353 | 0.884492 | 1.129987 |
| GnegativeGO | **0.154676** | **0.154676** | 0.215827 | 0.210432 | 0.232014 |
| GpositiveGO | 0.144231 | 0.144231 | 0.139423 | 0.129808 | **0.125000** |
| HumanGO | **0.713826** | 0.729904 | 0.877010 | 0.747588 | 1.143087 |
| PlantGO | 0.546154 | 0.546154 | **0.525641** | 0.633333 | 0.556410 |
| VirusGo | **0.373494** | **0.373494** | 0.457831 | 0.530120 | 0.457831 |
| EukaryotePseAAC | **2.776062** | 2.894466 | 3.068533 | 3.773166 | 3.075290 |
| GnegativePseAAC | **0.848921** | 0.866906 | 1.062950 | 2.568345 | 1.181655 |
| GpositivePseAAC | **0.653846** | 0.682692 | 0.918269 | 1.173077 | 0.923077 |
| HumanPseAAC | **2.438907** | 2.602090 | 2.657556 | 2.881029 | 2.717846 |
| Average | **0.928171** | 0.962621 | 1.081139 | 1.353139 | 1.154220 |

The best results are highlighted in bold.

TABLE V

PERFORMANCE COMPARISON OF GRRO-JS ALGORITHM AND OTHER
ALGORITHMS ON AVERAGE PRECISION

| | Average Precision | | | | |
|---|---|---|---|---|---|
| Datasets | GRRO-JS | GRRO | PMU | MDFS | D2F |
| EukaryoteGO | **0.888833** | **0.888833** | 0.864452 | 0.869276 | 0.841387 |
| GnegativeGO | **0.970671** | **0.970671** | 0.960552 | 0.955353 | 0.961065 |
| GpositiveGO | 0.947516 | 0.947516 | 0.947917 | 0.953125 | **0.954728** |
| HumanGO | **0.866900** | 0.866251 | 0.837906 | 0.863344 | 0.805080 |
| PlantGO | **0.876614** | **0.876614** | 0.873049 | 0.862683 | 0.862356 |
| VirusGo | **0.933133** | **0.933133** | 0.919478 | 0.902811 | 0.923494 |
| EukaryotePseAAC | **0.572548** | 0.554071 | 0.530864 | 0.403901 | 0.530712 |
| GnegativePseAAC | 0.770131 | **0.771615** | 0.724480 | 0.360723 | 0.702032 |
| GpositivePseAAC | **0.756811** | 0.745192 | 0.644231 | 0.553285 | 0.649840 |
| HumanPseAAC | **0.575082** | 0.539849 | 0.522005 | 0.511630 | 0.519622 |
| Average | **0.815824** | 0.809374 | 0.782493 | 0.723613 | 0.775031 |

The best results are highlighted in bold.

TABLE VI

PERFORMANCE COMPARISON OF GRRO-JS ALGORITHM AND OTHER
ALGORITHMS ON MACRO-F1

| | Macro-F1 | | | | |
|---|---|---|---|---|---|
| Datasets | GRRO-JS | GRRO | PMU | MDFS | D2F |
| EukaryoteGO | **0.530511** | **0.530511** | 0.402418 | 0.376189 | 0.327499 |
| GnegativeGO | **0.787441** | **0.787441** | 0.748347 | 0.665250 | 0.675818 |
| GpositiveGO | 0.780734 | 0.780734 | 0.727052 | 0.745140 | **0.834540** |
| HumanGO | 0.524492 | **0.528244** | 0.411520 | 0.485860 | 0.354071 |
| PlantGO | **0.600824** | **0.600824** | 0.552985 | 0.551853 | 0.529492 |
| VirusGo | **0.679621** | **0.679621** | 0.495093 | 0.631810 | 0.491611 |
| EukaryotePseAAC | 0.973666 | 0.972764 | 0.971788 | **0.974524** | 0.971642 |
| GnegativePseAAC | **0.986613** | 0.986573 | 0.982758 | 0.972093 | 0.983044 |
| GpositivePseAAC | **0.985036** | 0.984518 | 0.973586 | 0.978697 | 0.973967 |
| HumanPseAAC | 0.971700 | **0.972010** | 0.971897 | 0.971703 | 0.971861 |
| Average | 0.782064 | **0.782324** | 0.723744 | 0.735312 | 0.711355 |

The best results are highlighted in bold.

TABLE VII

PERFORMANCE COMPARISON OF GRRO-JS ALGORITHM AND OTHER
ALGORITHMS ON MICRO-F1

| | Micro-F1 | | | | |
|---|---|---|---|---|---|
| Datasets | GRRO-JS | GRRO | PMU | MDFS | D2F |
| EukaryoteGO | **0.812108** | **0.812108** | 0.782005 | 0.782099 | 0.748704 |
| GnegativeGO | **0.922535** | **0.922535** | 0.919298 | 0.896000 | 0.908289 |
| GpositiveGO | **0.914842** | **0.914842** | 0.901679 | 0.913462 | 0.911695 |
| HumanGO | 0.717546 | 0.720610 | 0.708731 | **0.736617** | 0.622900 |
| PlantGO | **0.756066** | **0.756066** | 0.756005 | 0.722295 | 0.739130 |
| VirusGo | **0.846939** | **0.846939** | 0.797753 | 0.825397 | 0.795455 |
| EukaryotePseAAC | **0.975593** | 0.974718 | 0.973752 | 0.975514 | 0.973628 |
| GnegativePseAAC | **0.987380** | 0.987335 | 0.984156 | 0.975416 | 0.984747 |
| GpositivePseAAC | **0.987228** | 0.986595 | 0.977818 | 0.981554 | 0.978141 |
| HumanPseAAC | 0.973796 | 0.974068 | 0.974025 | **0.974377** | 0.974129 |
| Average | 0.889403 | **0.889582** | 0.877522 | 0.878273 | 0.863682 |

The best results are highlighted in bold.

Compared to other algorithms, the GRRO and GRRO-JS algorithms generally performed better across the six evaluation metrics. Although the specific performance metrics varied significantly across different datasets, the overall trends among the algorithms remained consistent. For instance, the GRRO-JS algorithm consistently produced better evaluation results on the EukaryoteGO, GnegativeGO, and GnegativePseAAC datasets. However, on the GpositiveGO dataset, the evaluation results of the GRRO-JS algorithm were generally unsatisfactory, which may be related to the appropriateness of the dataset labeling or the suitability of the feature selection. Additionally, it can be observed that across these six evaluation metrics, the GRRO-JS algorithm outperformed most datasets. The average

values further highlight that the GRRO-JS algorithm exhibits superior average performance and generalization capability.

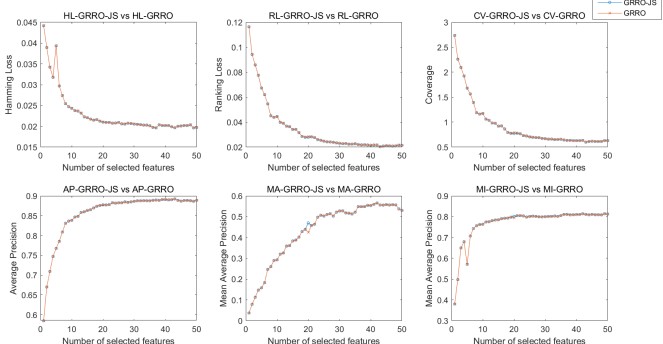

Fig. 2. Comparison of performance between GRRO-JS and GRRO algorithms on the EukaryoteGO dataset.

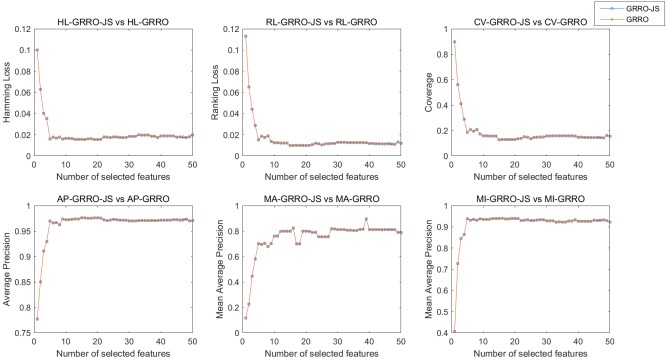

Fig. 3. Comparison of performance between GRRO-JS and GRRO algorithms on the GnegativeGO dataset.

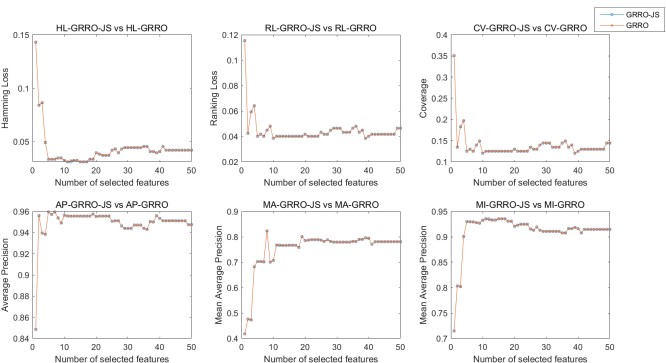

Fig. 4. Comparison of performance between GRRO-JS and GRRO algorithms on the GpositiveGO dataset.

*2) Comparison of the Performance Between the GRRO-JS Algorithm and the GRRO Algorithm:* Although GRRO-JS achieved better performance than GRRO on multiple datasets, the numerical differences between the two are very close. To further compare their performance, we conducted a more detailed comparison of the selected ten high-dimensional biological multi-label datasets. To distinguish the differences between

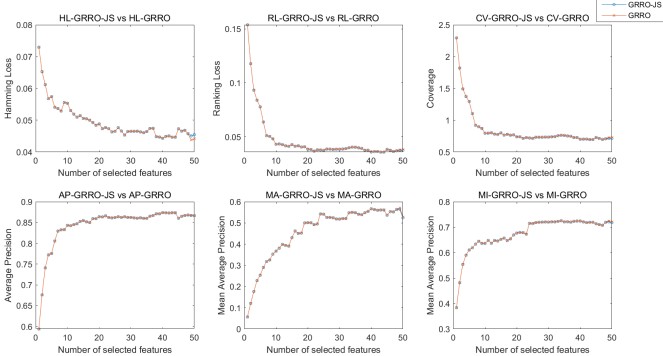

Fig. 5. Comparison of performance between GRRO-JS and GRRO algorithms on the HumanGO dataset.

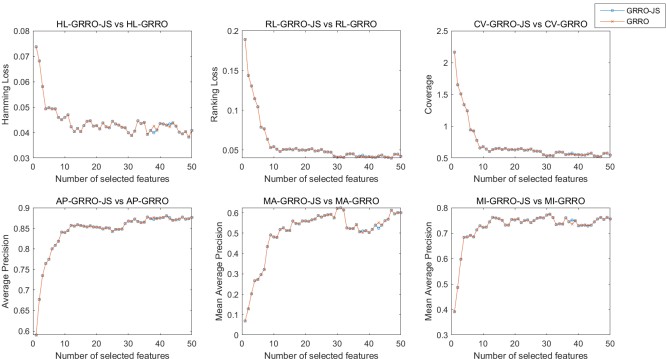

Fig. 6. Comparison of performance between GRRO-JS and GRRO algorithms on the PlantGO dataset.

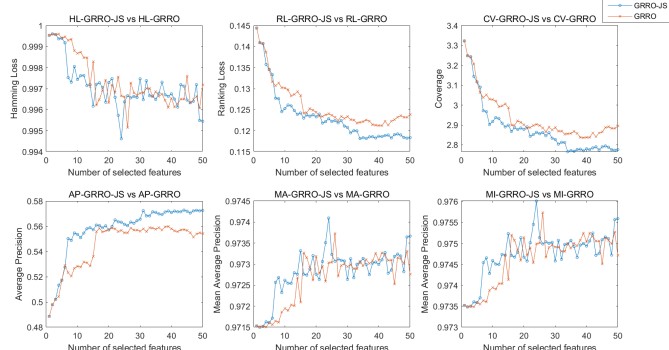

Fig. 8. Comparison of performance between GRRO-JS and GRRO algorithms on the EukaryotePseAAC dataset.

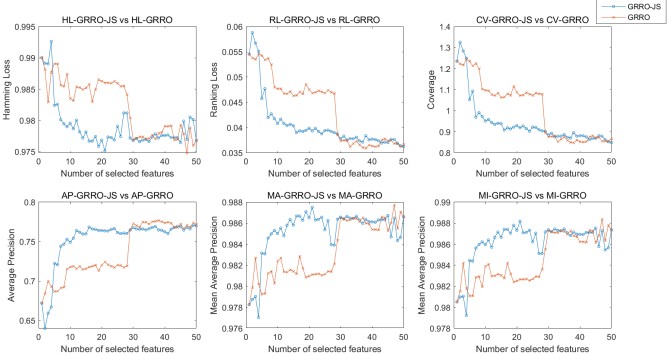

Fig. 9. Comparison of performance between GRRO-JS and GRRO algorithms on the GnegativePseAAC dataset.

the two more precisely, we evaluated their performance by selecting the top 1 to top 50 ranked features for each dataset, respectively.

The features of EukaryoteGO, GnegativeGO, GpositiveGO, HumanGO, PlantGO, and VirusGO datasets are all binary (0/1). Fig. 2 to Fig.7 show that the classification performance of GRRO and GRRO-JS algorithms on these high-dimensional biological information datasets is very similar. As seen in Fig. 2, GRRO and GRRO-JS achieved good classification results on the EukaryoteGO dataset for the multi-label classification of

eukaryotes. Although there are two points of sudden change in Hamming Loss and Micro F1, overall, the performance metrics gradually improve and converge as the number of selected features increases. Similar characteristics can be observed in the HumanGO and PlantGO datasets.

In contrast, the performance metrics of both algorithms on the GnegativeGO, GpositiveGO, and VirusGo datasets also improve as the number of selected features increases. Still, they exhibit two distinct phases: a sudden increase and stabilization. For example, as shown in Fig. 2 to Fig. 7, when

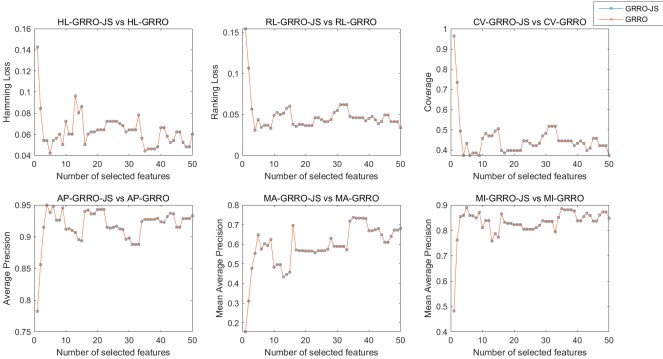

Fig. 7. Comparison of performance between GRRO-JS and GRRO algorithms on the VirusGO dataset.

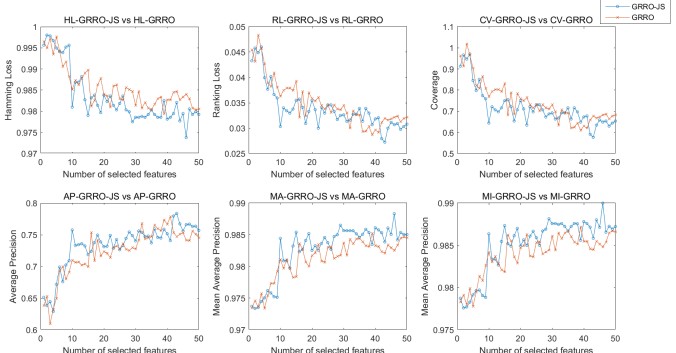

Fig. 10. Comparison of performance between GRRO-JS and GRRO algorithms on the GpositivePseAAC dataset.

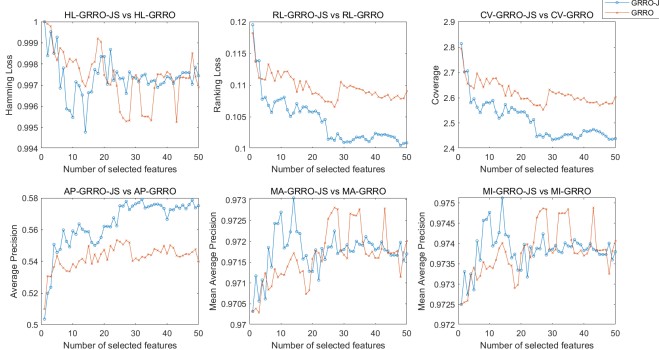

Fig. 11. Comparison of performance between GRRO-JS and GRRO algorithms on the HumanPseAAC dataset.

the number of selected features is less than 5, the Hamming Loss, Ranking Loss, and Coverage of the GpositiveGO dataset decrease rapidly as the number of features increases, while the Average Precision, Macro F1, and Micro F1 increase rapidly. After selecting more than five features, the values of the performance metrics gradually converge. We can observe that the GRRO-JS algorithm's performance is similar to that of the GRRO algorithm when handling multi-label classification tasks on high-dimensional biological data with binary features.

EukaryotePseAAC, GnegativePseAAC, GpositivePseAAC, and HumanPseAAC have continuous variable feature values. Fig. 8 to Fig. 11, respectively, illustrate the classification performance of the GRRO and GRRO-JS algorithms on these datasets.

As shown in Fig. 8 to Fig. 11, GRRO-JS achieves a significant advantage in all six performance metrics across the four datasets when dealing with continuous features. Taking the HumanPseAAC dataset as an example, GRRO-JS consistently outperformed the GRRO algorithm in ranking loss and coverage, with its scores almost always lower than those of GRRO. Similarly, in terms of average precision, GRRO-JS had consistently higher scores than GRRO, demonstrating a clear advantage. Both algorithms' performance scores gradually converged as the number of selected features increased in these three metrics. In Hamming loss, Macro F1, and Micro F1, GRRO-JS outperformed GRRO, although these three metrics did not show a convergence trend.

*3) Parameter Sensitivity Analysis of the GRRO-JS Algorithm:* We use the parameters $a$ and $beta$, conducting a grid search over the values $[10^{-3}, 10^{-2}, 10^{-1}, 10^0, 10^1, 10^2, 10^3]$. The ML-KNN classifier is employed for performance evaluation. The optimal parameters are determined by minimizing the average classification result (ACR) on the PlantGO dataset. The formula for ACR is given as $ACR(para) = \sum_{i=1}^{50} (HL_i(f,u) + RL_i(f,u))$, where $para$ represents the set of algorithm parameters, $u$ represents the test set, and $f$ represents the classifier. $HL_i(f,u)$ and $RL_i(f,u)$ are the sums of Hamming Loss and Ranking Loss, respectively, when selecting the top i features. The parameter sensitivity analysis for $a$ and $beta$ is shown in Fig. 12. As the figure shows, when $a$

is set to 100 and $beta$ to 0.1, the GRRO-JS algorithm achieves the best performance.

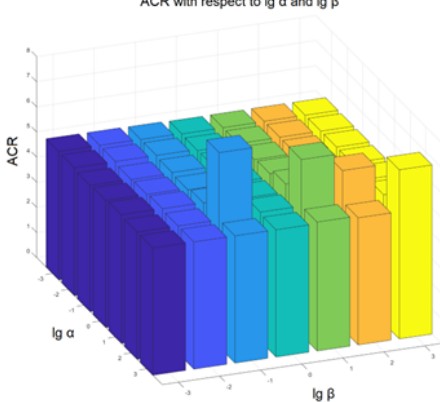

Fig. 12. Sensitivity analysis of parameters $a$ and $\beta$.

## V. CONCLUSION

Building on the GRRO algorithm, we introduce JS divergence to measure the correlation between labels. Additionally, we implement data discretization preprocessing for continuous datasets. Furthermore, we employ parallel computing for correlation measurement, significantly reducing the algorithm's time complexity. We conduct a detailed analysis and formula derivation of the GRRO algorithm. By incorporating JS divergence, the GRRO-JS algorithm optimizes the handling of label correlations, enhancing the accuracy and computation speed of the GRRO algorithm. Experimental results on ten high-dimensional biological datasets, evaluated using six performance metrics, demonstrate that the GRRO-JS algorithm improves the accuracy and efficiency of feature selection.

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
