# OpenReview forum: "Multi-Label Feature Selection for High-Dimensional Biological Data via Global Relevance and Redundancy Optimization based on JS Divergence"
_IEEE.org/ICIST/2024/Conference — IEEE ICIST 2024 Conference Submission_

### Official Review · Reviewer_o4yw · 2024-08-21
**Multi-Label Feature Selection for High-Dimensional Biological Data via Global Relevance and Redundancy Optimization based on JS Divergence**

**Rating:** 7
**Confidence:** 2

**Review:**

Building on the GRRO algorithm, the authors introduce JS divergence to measure the correlation between labels. Additionally, the authors implement data discretization preprocessing for continuous datasets. Furthermore, we employ parallel computing for correlation measurement, significantly reducing the algorithm’s time complexity. This paper is well organized. However, the figures in the simulation are unclear, and the authors need to adjust them appropriately. In addition, research motivation should be emphasized.

---

### Official Review · Reviewer_kmoP · 2024-08-21
**Accept**

**Rating:** 7
**Confidence:** 5

**Review:**

The paper introduces JS divergence into the GRRO algorithm to quantify the correlation between labels and incorporates a data discretization preprocessing step for continuous datasets,underscoring its practical significance.A discussion of the algorithm's time complexity and potential optimizations could benefit readers interested in implementation efficiency.The work is innovative and practical,however,a detailed analysis of the algorithm's time and space complexity could be included to better understand its performance.

---

### Official Review · Reviewer_CUSd · 2024-08-22
**Accept**

**Rating:** 8
**Confidence:** 4

**Review:**

This paper proposes the GRRO-JS algorithm which combines JS divergence and the GRRO  (Global Relevance and Redundancy Optimization) algorithm to measure the correlation between labels and introduces data discretization pre-processing operations for continuous data sets and effectively eliminate the interference of abnormal data. This paper employs parallel computing techniques to measure the correlation between features, significantly reducing the algorithm’s time complexity. Ultimately, the experimental results in this paper demonstrate the superiority of GRRO-JS in terms of both computational efficiency and accuracy. There are the following suggestions for this paper:
1. The abstract contains some elements that are redundant, such as the application of  multi-label feature selection. Therefore, it is recommended that the abstract could be simplified.
2. The interpretation of originality could be made clearer.

---

### Decision · Program_Chairs · 2024-09-08

Accept (Oral)